# A Robust Fault-Tolerant Predictive Control for Discrete-Time Linear Systems Subject to Sensor and Actuator Faults

**DOI:** 10.3390/s21072307

**Published:** 2021-03-25

**Authors:** Sofiane Bououden, Ilyes Boulkaibet, Mohammed Chadli, Abdelaziz Abboudi

**Affiliations:** 1Laboratory of SATIT, Department of Industrial Engineering, University Abbes Laghrour, Khenchela 40004, Algeria; sofiane.bououden@univ-khenchela.dz; 2College of Engineering and Technology, American University of the Middle East, Egaila 54200, Kuwait; Ilyes.boulkaibet@aum.edu.kw; 3Université Paris-Saclay, Univ Evry, IBISC, 91020 Evry, France; 4Department of Mechanical Engineering, Faculty of Sciences and Technology, University Abbes Laghrour, Khenchela 40004, Algeria; aboudiabdelaziz23@gmail.com

**Keywords:** fault-tolerant predictive control, observer-based control, actuator and sensor faults, input constraints, linear matrix inequalities (LMIs)

## Abstract

In this paper, a robust fault-tolerant model predictive control (RFTPC) approach is proposed for discrete-time linear systems subject to sensor and actuator faults, disturbances, and input constraints. In this approach, a virtual observer is first considered to improve the observation accuracy as well as reduce fault effects on the system. Then, a real observer is established based on the proposed virtual observer, since the performance of virtual observers is limited due to the presence of unmeasurable information in the system. Based on the estimated information obtained by the observers, a robust fault-tolerant model predictive control is synthesized and used to control discrete-time systems subject to sensor and actuator faults, disturbances, and input constraints. Additionally, an optimized cost function is employed in the RFTPC design to guarantee robust stability as well as the rejection of bounded disturbances for the discrete-time system with sensor and actuator faults. Furthermore, a linear matrix inequality (LMI) approach is used to propose sufficient stability conditions that ensure and guarantee the robust stability of the whole closed-loop system composed of the states and the estimation error of the system dynamics. As a result, the entire control problem is formulated as an LMI problem, and the gains of both observer and robust fault-tolerant model predictive controller are obtained by solving the linear matrix inequalities (LMIs). Finally, the efficiency of the proposed RFTPC controller is tested by simulating a numerical example where the simulation results demonstrate the applicability of the proposed method in dealing with linear systems subject to faults in both actuators and sensors.

## 1. Introduction

Generally, the reliability of control systems, in terms of the stability of the entire system and uncertainty rejection, has become an important subject in the last three decades where the complexity of control systems has increased significantly due to the fast technological development in many engineering areas. Usually, industrial plants and systems rely on measurements obtained by sensors as well the control actions from actuators to achieve certain control objectives, which implies that the quality of the sensor and actuator measurements have significant impacts on the control system performance. As a result, the controlling process will eventually become more difficult if faults or failures occur in system measurements. Faults in the system may appear from different sources, such as faults caused by sensors due to aging or broken components. Additionally, faults can also be caused by actuators, such as the presence of broken parts, certain failures in the actuator machinery, or errors in the operating mechanism. As a result, the presence of faults and failures within a system might weaken the system’s performance and lead to system degradation or stability loss.

Generally, both actuators and sensors may partially or completely lose effectiveness due to several causes resulting in unexpected faults in measurements or control actions. Furthermore, actuators and sensors may also provide unwanted offsets and becomes stuck during operation. Furthermore, faults caused by sensors and actuators may accumulate with time and increase after each closed-loop process, causing significant damage to the entire system. Therefore, it is necessary to build advanced fault detection and isolation (FDI) techniques to estimate faults in the system, and synthesize fault-tolerant control (FTC) methods in order to augment the system, and guarantee the reliability, sustainability, and safety of the systems [1,2,3,4].

Over the past three decades, various researchers and practitioners have been involved in developing and improving fault-tolerant control (FTC) approaches, where several approaches have been proposed and used to enhance the controlling process under the presence of sensors and actuator faults. Generally, FTC methods are divided into two different categories: the first class is the passive FTC, while the second class is the active FTC method. In the passive FTC category, the control system does not instantly respond to the online changes or occurrence of faults, while the controller is pre-designed, and its structure is fixed for both the faulty-free and the faulty cases. As a result, the control system will only exhibit certain robustness and tolerate a set of faults without any change in the structure of the controller [5,6,7]. Consequently, this type of controller is easy to implement. However, it may display limited performance and small fault tolerance, since the controller structure cannot adjust to new online faults information. On the other hand, the structure of an active FTC allows the controller to react to the occurrence of faults immediately and change its parameters accordingly in order to achieve robustness and tolerate faults based on online faults information obtained by the fault detection and identification (FDI) procedure [8,9,10]. In conclusion, active FTC schemes may have the ability to deal with online faults and can achieve better robustness with significant fault acceptability than the passive approach [10].

Usually, noises and external disturbances are undesired factors in real-word control systems, and these unavoidable disturbances can be generated by many sources including the dynamics of the targeted system itself. In order to achieve a better performance, it is essential to reduce the impact of noises and uncertainties to a certain adequate level. Furthermore, having enough information of all states of a system is a very complex task in many industrial engineering systems where some of these states are unmeasurable. To deal with this issue, observers and observer-based controllers are employed to control these types of systems, where an observer has the ability to estimate the states of a dynamic system. Consequently, the observer provides important information, which eventually is used for system supervision and the realization of the feedback control [11,12,13,14,15,16]. Many controllers based on observer have been proposed to control systems with unmeasurable states. Dong et al. [17] proposed an approach based on H-infinity in a generalized internal model control design with an adaptive observer-based fault estimator for active fault-tolerant control. The proposed approach uses online fault estimations to guarantee H-infinity performance and closed-loop system stability. Aouaouda et al. [18] proposed an active fault-tolerant tracking controller methodology to deal with vehicle dynamics. In this approach, a Takagi–Sugeno model was used to represent the dynamics of the nonlinear system, while the system states along with sensor faults are estimated by a descriptor observer. Finally, a fault-tolerant tracking controller is proposed in terms of linear matrix inequalities (LMIs) to control vehicle dynamics. Kommuri et al. [1] proposed a fault-tolerant control approach based on a higher-order sliding mode observer. The proposed approach was investigated in controlling an automatic speed tracking control of an electric vehicle powered by a permanent-magnet synchronous motor. Furthermore, the proposed approach was examined in the presence of sensor failures/faults. Unfortunately, the nature of the proposed observer is very complex, and the controller gains were set beforehand. Additionally, the proposed approach in [1] considered a Lyapunov matrix in a specific form. The authors in [19,20] proposed fault-tolerant controlling strategies based on H-infinity observers to estimate both system states and faults. These types of observers have the ability to estimate both states and faults with high accuracy. Based on the proposed observer, output feedback fault-tolerant controllers were synthesized to ensure the stability of the targeted systems. Furthermore, several novel observer designs have been proposed for discrete-time systems. Most recently, a novel observer design approach was presented in [21], in which a robust stable hybrid fault-tolerant predictive control was proposed for handling actuator faults providing effective robust trajectory tracking performance.

As one of the most popular advanced control algorithms in the industry, model predictive control (MPC) has been introduced into several industrial processes and become one of the hottest topics in research and development [22,23,24,25,26]. Recently, fault-tolerant control approaches based on model predictive control (MPC) have gained more attraction in several fields where several researchers have tried to address the faults/failures issues within the use of predictive control. Zou et al. [27] proposed a fault-tolerant model predictive control (MPFTC) strategy for the purpose of controlling batch processes subject to disturbances and partial actuator faults. In this approach, the control procedure was formulated as an optimization problem, while a genetic algorithm was used to obtain the control signal to enhance the performance of the overall system. Zarch et al. [28] employed the viability theory to evaluate actuator fault tolerance based on nonlinear model predictive control (NMPC). In this approach, the nonlinear model of the system was described as a linear parameter varying (LPV) form, while the viability concepts were applied to evaluate the fault tolerance of the NMPC strategy. Shi et al. [29] proposed a robust model predictive control for a class of industrial processes with interval time-varying delay subject to constraints, uncertainties, unknown disturbances, and partial actuator failures. In this approach, an extended state-space model was used to describe the system, while this extended model was used to synthesize a robust control low that guarantees acceptable tracking performance. Sheikhbahaei et al. [30] constructed a novel approach for constrained explicit model predictive control for linear discrete-time systems subject to uncertainties, disturbances, and possible actuator faults. Zhang et al. [31] developed a model predictive fault-tolerant control approach for batch processes subject to partial actuator faults and unknown disturbances, while in [32], an overview of predictive control for time-delay systems is proposed.

Unfortunately, most of the existing fault tolerance predictive control approaches for discrete-time systems are either focused only on partially actuator faults or sensor faults, and a simultaneous sensor/actuator fault problem has not been fully studied for discrete-time systems. Motivated by the results provided in [19,20], this manuscript is dedicated to developing and investigating a novel robust constrained fault tolerant predictive control method for discrete-time systems subject to disturbances and both sensor and actuator faults (see Figure 1). Consider the proposed discrete-time control scenario shown in Figure 1. The measurements are obtained by sensors, and used by both virtual and real observers to estimate the system’s states. The estimated states are then used to compute the desired control signal. To construct the control low, an extended state-space model is first used for constructing virtual observers. Then, real observers are formulated based on virtual observers. This step is quite obvious, since virtual observers may contain unavailable information. Finally, an observer-based robust fault tolerance predictive control strategy is designed such that both system states and actuator/sensor faults are estimated at once. As a result, these steps can ensure a minor influence of faults on the dynamics of the system. Furthermore, the robust stability of the closed-looped system is guaranteed with several proposed stability conditions. As a result, these proposed sufficient stability conditions confirm the robust stability of the overall closed-loop system, while both observer and fault-tolerant controller gains are obtained by solving the linear matrix inequalities (LMIs) problem. Note that the robust fault-tolerant predictive control, which is an observer-based controller, is synthesized via an online LMI optimization problem to minimize the upper bound of the cost function while handling disturbances, faults, and constraints.

This manuscript is structured as follows: First, the problem formulation is presented in Section 2. Section 3 describes the steps of constructing the virtual and real observers in order to estimate the states and faults at once. Next, the proposed robust constrained fault tolerant predictive control is described in detail in Section 4. Section 5 is dedicated to proving the robust stability of the proposed RFTPC scheme. Then, simulation results are illustrated in Section 6. Finally, Section 6 concludes the paper.

Notation: all symbols used in this paper are standard, in the area of robust control and fault-tolerant control, unless otherwise specified. Given a vector s, the symbol ‖s‖ signifies the Euclidean norm of vector s. K is the controller gain matrices, while L represents the observer gain matrix. ℝn represents the n dimensional Euclidean space. 0n and I represent the zero and the identity matrices with proper dimensions, respectively. The superscript T, usually seen in the top-right of matrices and vectors, represents the transpose of matrices or vectors. Finally, matrices X, W, G, Y, M, and H are symmetric positive definite matrices. Q and R are known as positive-definite weighting matrices.

## 2. Problem Statement

Generally, a discrete-time system can be represented as follow:(1){x(k+1)=Ax(k)+Bu(k)y(k)=Cx(k)
where x(k)∈ℝnx is the state vector of the system, y(k)∈ℝny represents the output vector of the system, and u(k)∈ℝnu is the system input vector.

The control signal is usually governed by the following constraints:(2)$${\left\|u\left(k\right)\right\|}_{2}\leq u_{max}$$
and $u_{max}$ is the upper limit on the control signal.

Since the considered discrete-time system is subject to both disturbances and faults, an observer that can be used to estimate system states and faults with high accuracy is very hard to attain. Thus, an extra step based on virtual observers is considered in order to design the real observer. Generally, both sensor and actuator faults are considered in this work. The real measurement of the system output is usually altered when a sensor fault occurs and modelled as follows:(3)$$y\left(k\right)=Cx\left(k\right)+f_{s}\left(k\right)$$

fs∈ℝny represents the vector of additive sensor faults.

Similarly, an actuator fault can cause certain changes over the control input, which can be represented as:(4)$$u\left(k\right)=u_{c}\left(k\right)+f_{a}\left(k\right)$$
where $f_{a}\left(k\right)$ describes the actuator fault, and $u_{c}\left(k\right)$ is the actual control input signal.

Through this work, several Lemmas and assumptions are required for the development of our main results.

**Lemma** **1.**
*(Schur complement). [33] Given any real matrices,*
X
*,*
Y
*, and*
Z
*with*
$X=X^{T}$
*and*
Z>0
*. Then, we have the following:*
$$X-YZ^{-1}Y^{T}<0$$


If and only if
$$\left[\begin{array}{cc} X & Y^{T}\\ Y & Z \end{array}\right]<0$$

**Assumption** **1.***Actuator faults*$f_{a}\left(k\right)$*, sensor faults*$f_{s}\left(k\right)$*, and the external disturbance*ω(k)*are bounded, and there is a positive scalar*β*such that*‖ω(k)‖<β.

**Remark** **1.**
*Assumption 1 will eventually ensure that the increments of the disturbances as well as faults between two sampling time instants are bounded.*


In this paper, both sensor and actuator faults are considered and added to the model representation in Equation (1) as follows:(5){x(k+1)=Ax(k)+B(u(k)+fa(k))+Dω(k)yf(k)=Cx(k)+fs(k)
where x∈ℝnx, uf∈ℝnu, and yf∈ℝny denote the vector of the faulty system state, faulty control input, and faulty measurement output vector, respectively. The vectors fa∈ℝnu and fs∈ℝny are the additive actuator and sensor faults, respectively. A, B, C, and D are the system matrices of compatible dimensions.

## 3. Observer Design

The main objective of a virtual observer is to improve measurements and provide better accuracy in estimating system states and fault observation.

To design an observer that estimates the system state vector x(k), sensor faults $f_{s}\left(k\right)$, and actuator faults $f_{a}\left(k\right)$, an augmented system based on the system Equation (5) and subject to external disturbance is defined as:(6)$$\begin{array}{cc} E_{1}𝓏\left(k+1\right) & =A𝓏\left(k\right)+Bu_{c}\left(k\right)+D\omega \left(k\right)\\ y_{f}\left(k\right) & =E_{2}𝓏\left(k\right) \end{array}$$
where the augmented vector $𝓏=\left[\begin{array}{c} x\left(k\right)\\ f_{s}\left(k\right)\\ f_{a}\left(k\right) \end{array}\right]$, and matrices $A=\left[\begin{array}{ccc} A & 0 & B \end{array}\right]$, $E_{1}=\left[\begin{array}{ccc} I & 0 & 0 \end{array}\right]$, $E_{2}=\left[\begin{array}{ccc} C & I & 0 \end{array}\right]$, and $E_{3}=\left[\begin{array}{ccc} 0 & 0 & I \end{array}\right]$.

Consider $E={\left[\begin{array}{ccc} {E_{1}}^{T} & {E_{2}}^{T} & {E_{3}}^{T} \end{array}\right]}^{T}=\left[\begin{array}{ccc} I & 0 & 0\\ C & I & 0\\ 0 & 0 & I \end{array}\right]$ a full rank matrix with I, and 0 represents the identity and the zero matrices with proper dimensions. $T=\left[\begin{array}{ccc} T_{1} & T_{2} & T_{3} \end{array}\right]$ denotes the inverse matrix of E, which can be easily obtained by $T=E^{-1}$, i.e., $T_{1}E_{1}+T_{2}E_{2}+T_{3}E_{3}=I$, and the vectors of matrix T are defined as $T=\left[\begin{array}{ccc} T_{1} & T_{2} & T_{3} \end{array}\right]=\left[\begin{array}{ccc} I & 0 & 0\\ -C & I & 0\\ 0 & 0 & I \end{array}\right]$.

Multiplying $T_{1}$ by both sides of Equation (6), and by using the fact $T_{1}E_{1}+T_{2}E_{2}+T_{3}E_{3}=I$, we have the following:(7)$$𝓏\left(k+1\right)=T_{1}A𝓏\left(k\right)+T_{1}Bu_{c}\left(k\right)+T_{1}D\omega \left(k\right)+\left(T_{2}E_{2}+T_{3}E_{3}\right)𝓏\left(k+1\right)$$

Defining a virtual observer in the following form:(8)$$\hat{𝓏}\left(k+1\right)=T_{1}A\hat{𝓏}\left(k\right)+T_{1}Bu_{c}\left(k\right)+T_{1}D\omega \left(k\right)+T_{2}E_{2}𝓏\left(k+1\right)+L\left(y_{f}\left(k\right)-E_{2}\hat{𝓏}\left(k\right)\right)$$
where $\hat{𝓏}=\left[\begin{array}{c} \hat{x}\left(k\right)\\ {\hat{f}}_{s}\left(k\right)\\ {\hat{f}}_{a}\left(k\right) \end{array}\right]$ represents the estimated values of the system state vector x(k), sensor faults $f_{s}\left(k\right)$, and actuator faults $f_{a}\left(k\right)$, respectively. Finally, L is the targeted observer gain matrix.

Next, the estimation error e(k) of the observer in Equation (8) is defined as $e\left(k\right)=𝓏\left(k\right)-\hat{𝓏}\left(k\right)$, and by subtracting Equation (8) from Equation (7), the dynamic equation of error is obtained as follows:(9)$$\begin{array}{c} e\left(k+1\right)=\left(T_{1}A-LE_{2}\right)e\left(k\right)+T_{1}D\omega \left(k\right)+T_{3}E_{3}𝓏\left(k+1\right)\\ e\left(k+1\right)=\left(T_{1}A-LE_{2}\right)e\left(k\right)+D\bar{\omega }\left(k\right) \end{array}$$
where $D=\left[\begin{array}{cc} D & 0\\ -CD & I \end{array}\right]$, $\bar{\omega }\left(k\right)=\left[\begin{array}{c} \omega \left(k\right)\\ f_{a}\left(k+1\right) \end{array}\right]$.

Unfortunately, the virtual observer contains the unknown term $T_{2}E_{2}𝓏\left(k+1\right)$ in Equation (8), which cannot be easily determined and will eventually reduce the precision of the observed information. To address this issue, a real observer based on the previous virtual observer parameters is designed for eliminating the undesirable term $T_{2}E_{2}𝓏\left(k+1\right)$.

Therefore, an auxiliary variable $\vartheta \left(k\right)=\hat{𝓏}\left(k\right)-T_{2}E_{2}𝓏\left(k\right)$ is introduced, while the real observer of the system Equation (5) is obtained as follows:(10)$$\begin{array}{c} \vartheta \left(k+1\right)=\bar{A}\vartheta \left(k\right)+\bar{ℬ}u_{c}\left(k\right)+ℒ\hat{𝓏}\left(k\right)\\ \hat{𝓏}\left(k\right)=\vartheta \left(k\right)+\bar{C}y_{f}\left(k\right) \end{array}$$
where ϑ(k) is an auxiliary variable, and matrices $\bar{A}$, $\bar{ℬ}$, $\bar{C}$, and ℒ are the observer parameters to be determined later as follows:(11)$$\bar{A}=T_{1}A-LE_{2},\;\bar{ℬ}=T_{1}B,\;\bar{C}=T_{2}E_{2},\;\mathrm{and}\;ℒ=L+\left(T_{1}A-LE_{2}\right)T_{2}$$

## 4. Observer-Based Robust Tolerant Predictive Control

In this section, an observer-based state feedback fault-tolerant predictive controller was constructed for stabling the overall closed-loop system. First, an observer was constructed in order to estimate faults and in the presence of external disturbance. Next, a fault-tolerant predictive controller based on observers was designed to ensure the stability of the system and maintained acceptable control performance.

As the first step, the control law for the system Equation (2) was designed to guarantee better fault compensation and reject the influences of disturbances. The control signal is considered in the following form:(12)$$u_{c}\left(k\right)=KE_{1}\hat{𝓏}\left(k\right)-E_{3}\hat{𝓏}\left(k\right)$$

Substituting $u_{c}\left(k\right)$ it into Equation (4), we have the following:$$u\left(k\right)=KE_{1}\hat{𝓏}\left(k\right)-E_{3}\hat{𝓏}\left(k\right)+f_{a}\left(k\right)$$
$$u\left(k\right)=KE_{1}\hat{𝓏}\left(k\right)+KE_{1}𝓏\left(k\right)-KE_{1}𝓏\left(k\right)-E_{3}\hat{𝓏}\left(k\right)+E_{3}𝓏\left(k\right)-E_{3}𝓏\left(k\right)+f_{a}\left(k\right)$$
where $\left(KE_{1}-E_{3}\right)z\left(k\right)=Kx\left(k\right)+f_{a}\left(k\right)$, then the control signal is represented as:(13)$$u\left(k\right)=Kx\left(k\right)+\left(E_{3}-KE_{1}\right)e\left(k\right)$$

Replacing Equation (13) with Equation (5), we obtain the following:$$x\left(k+1\right)=Ax\left(k\right)+B\left(Kx\left(k\right)+\left(E_{3}-KE_{1}\right)e\left(k\right)+f_{a}\left(k\right)\right)+D\omega \left(k\right)$$
(14)$$x\left(k+1\right)=\left(A+BK\right)x\left(k\right)+B\left(E_{3}-KE_{1}\right)e\left(k\right)+\bar{D}\bar{\omega }\left(k\right)$$
where $\bar{D}=\left[\begin{array}{cc} D & 0 \end{array}\right]$, $\bar{\omega }\left(k\right)=\left[\begin{array}{c} \omega \left(k\right)\\ f_{a}\left(k+1\right) \end{array}\right]$.

Next, an optimization problem is established for obtaining a suitable control signal where the objective is to minimize a worst-case quadratic objective function in an infinite horizon as follows:(15)$$\begin{array}{c} \underset{u_{c}\left(k\right)}{\underbrace{\min}}\max J_{\infty }\left(k\right)\\ \mathrm{Subject}\;\mathrm{to}\;\left(7\right)\;\mathrm{and}\;{\left\|u\left(k\right)\right\|}_{2}\leq u_{max} \end{array}$$
(16)$$\mathrm{where}\;J_{\infty }\left(k\right)=\sum_{i=0}^{\infty }{\left\|x\left(k+i\right)\right\|}_{Q}^{2}+{\left\|u\left(k+i\right)\right\|}_{R}^{2}+\mu^{2}{\left\|\bar{\omega }\left(k\right)\right\|}^{2}$$

The matrices Q and R are known as positive-definite weighting matrices. Next, sufficient stability conditions for observation errors and an overall closed-loop system are established in terms of LMIs. Based on Kothare et al. [34], the following theorem is proposed for the robust predictive control.

**Theorem** **1.**
*Given an augmented state-space model described by Equation (6), the state-feedback controller described in Equation (4) will robustly stabilize the augmented system subjected to disturbances, sensor/actuator faults, and input constraints if there are symmetric positive definite matrices*
X
*,*
W
*,*
G
*,*
Y
*,*
M,
*and*
H
*and a positive scalar*
γ
*satisfying the following convex optimization problem:*
(17)$$\underset{\gamma ,X,\;W,\;G,\;Y,\;M,H}{\min}\gamma$$


Subject to
(18)$$\left[\begin{array}{ccc} -1 & * & *\\ x\left(k\right) & -X & *\\ e\left(k\right) & 0 & -W \end{array}\right]\leq 0$$
(19)$$\left[\begin{array}{cc} -u_{max}^{2} & Y\\ Y^{T} & X-G^{T}-G \end{array}\right]\leq 0$$
(20)$$\left[\begin{array}{cccccccc} -{Ф}_{1} & * & * & * & * & * & * & *\\ R^{\frac{1}{2}}Y & -\gamma I & * & * & * & * & * & *\\ Q^{\frac{1}{2}}G & 0_{n} & -\gamma I & * & * & * & * & *\\ 0 & 0_{n} & 0_{n} & -\frac{\mu^{2}I}{\gamma } & * & * & * & *\\ 0 & 0_{n} & 0_{n} & 0_{n} & -{Ф}_{2} & * & * & *\\ R^{\frac{1}{2}}\left(E_{3}G-YE_{1}\right) & 0_{n} & 0_{n} & 0_{n} & 0_{n} & -\gamma I & * & *\\ AG+BY & B\left(E_{3}G-YE_{1}\right) & \bar{D}G & 0_{n} & 0_{n} & 0_{n} & -X & *\\ 0_{n} & T_{1}MA-HE_{2} & MD & 0_{n} & 0_{n} & 0_{n} & 0_{n} & -W \end{array}\right]<$$
where ${Ф}_{1}=-X+G^{T}+G$ and ${Ф}_{2}=-W+G^{T}+G$.

Additionally, at each sampling time, the control gain can be easily obtained by $K=YX_{1}^{-1},$ while the gain of the state virtual observer by $L=M^{-1}H$. Furthermore, the obtained observer gain is used to attain the ℒ of the real observer in Equation (11).

**Proof.** To obtain the stability conditions, a Lyapunov function is defined as:
(21)$$V(k/k)=x^{T}(k/k)Sx(k/k)+e^{T}(k/k)Pe(k/k)$$
S>0 and P>0.And
(22)$$V\left(k+i/k\right)=x^{T}\left(k+i/k\right)Sx\left(k+i/k\right)+e^{T}\left(k+i/k\right)Pe\left(k+i/k\right)$$For any i≥0, suppose V(k+i/k) satisfies the following stability constraint:(23)$$V\left(k+i+1/k\right)-V\left(k+i/k\right)\leq -\left[{\left\|x\left(k+i/k\right)\right\|}_{Q}^{2}+{\left\|u\left(k+i/k\right)\right\|}_{R}^{2}\right]+\mu^{2}{\left\|\bar{\omega }\left(k\right)\right\|}^{2}$$Assumed that the summation is up to ∞, i.e., i→∞, x(∞)=0. Summing from i=0 to ∞ produces the following:(24)$$J_{\infty }\left(k\right)\leq V\left(k+i/k\right)$$By defining V(k+i/k)≤γ, an upper bound on the performance index is obtained as $J_{\infty }\left(k\right)\leq \gamma$. Hence, the first inequality of Equation (23) holds.Next, we show that the second inequality of Equation (24) holds.
(25)$$x^{T}\left(k/k\right)S\;x\left(k/k\right)+e^{T}\left(k/k\right)P\;e\left(k/k\right)\leq \gamma$$Since the inequality in Equation (20) implies that V(ζ(k+j+1/k)) strictly decreases as j transitions to ∞ and V(ζ(k/k))≤γ from Equation (21), we have the following:(26)$$\frac{1}{\gamma }x^{T}\left(k/k\right)S\;x\left(k/k\right)+\frac{1}{\gamma }e^{T}\left(k/k\right)P\;e\left(k/k\right)\leq 1$$Using the Schur complement, we obtain the following:(27)$$\left[\begin{array}{ccc} -1 & * & *\\ x\left(k\right) & -\gamma S^{-1} & *\\ e\left(k\right) & 0 & -\gamma P^{-1} \end{array}\right]\leq 0$$Substituting $S=\gamma X^{-1}$ and $P=\gamma W^{-1}$ into the above inequality matrix, and then applying the congruence transformation to the resulting inequality with diag [1, $X^{-1}$, $W^{-1}$], we conclude that Equation (18) holds.Based on input constraint Equation (2), we can write the following:(28)$${\left\|u\left(k+i/k\right)\right\|}_{max}≜\underset{i}{max}u_{i}\left(k+i/k\right)$$
(29)$$\underset{i>0}{max}{\left\|u\left(k\right)\right\|}_{max}=\underset{i>0}{max}{\left\|K\hat{x}\left(k\right)\right\|}_{max}$$
(30)$$\underset{i>0}{max}{\left\|u\left(k\right)\right\|}_{max}\leq {\left\|K{\left(\frac{1}{\gamma }X\right)}^{\frac{1}{2}}\right\|}_{2}^{2}$$
and based on Equation (30), we can obtain the following:(31)$$u_{max}^{2}\leq {\left(\frac{1}{\gamma }X\right)}^{\frac{1}{2}T}K^{T}K{\left(\frac{1}{\gamma }X\right)}^{\frac{1}{2}}$$
(32)$$-u_{max}^{2}+{\left(\frac{1}{\gamma }X\right)}^{\frac{1}{2}T}K^{T}K{\left(\frac{1}{\gamma }X\right)}^{\frac{1}{2}}\leq 0$$Using the Schur complement, we obtain the following:(33)$$\left[\begin{array}{cc} -u_{max}^{2} & K\\ K^{T} & -X^{-1} \end{array}\right]\leq 0$$Multiplying the right by $\left[\begin{array}{cc} I & O\\ 0 & G \end{array}\right]$ and the left by $\left[\begin{array}{cc} I & O\\ 0 & G^{T} \end{array}\right],$ we obtain Equation (19).Finally, Equation (20) implies the following:(34)$$\left[\begin{array}{cccccccc} -{Ф}_{1} & * & * & * & * & * & * & *\\ R^{\frac{1}{2}}Y & -\gamma I & * & * & * & * & * & *\\ Q^{\frac{1}{2}}G & 0_{n} & -\gamma I & * & * & * & * & *\\ 0 & 0_{n} & 0_{n} & -\frac{\mu^{2}I}{\gamma } & * & * & * & *\\ 0 & 0_{n} & 0_{n} & 0_{n} & -{Ф}_{2} & * & * & *\\ R^{\frac{1}{2}}\left(E_{3}G-YE_{1}\right) & 0_{n} & 0_{n} & 0_{n} & 0_{n} & -\gamma I & * & *\\ AG+BY & B\left(E_{3}G-YE_{1}\right) & \bar{D}G & 0_{n} & 0_{n} & 0_{n} & -X & *\\ 0_{n} & T_{1}MA-HE_{2} & MD & 0_{n} & 0_{n} & 0_{n} & 0_{n} & -W \end{array}\right]<0$$
where ${Ф}_{1}=-X+G^{T}+G$, ${Ф}_{2}=-W+G^{T}+G,$ and the matrices X and W are positives. Furthermore, since X>0, we have the following:(35)$${\left(G^{T}-X\right)}^{T}X^{-1}\left(G-X\right)\geq 0\Rightarrow -X+G^{T}+G\leq G^{T}X^{-1}G$$Therefore,
(36)$$-W+G^{T}+G\leq G^{T}W^{-1}G$$By taking into account Equations (35) and (36), we can state that Equation (20) implies the following:(37)$$\left[\begin{array}{cccccccc} -G^{T}X^{-1}G & * & * & * & * & * & * & *\\ R^{\frac{1}{2}}Y & -\gamma I & * & * & * & * & * & *\\ Q^{\frac{1}{2}}G & 0_{n} & -\gamma I & * & * & * & * & *\\ 0 & 0_{n} & 0_{n} & -\frac{\mu^{2}I}{\gamma } & * & * & * & *\\ 0 & 0_{n} & 0_{n} & 0_{n} & -G^{T}W^{-1}G & * & * & *\\ R^{\frac{1}{2}}\left(E_{3}G-YE_{1}\right) & 0_{n} & 0_{n} & 0_{n} & 0_{n} & -\gamma I & * & *\\ AG+BY & B\left(E_{3}G-YE_{1}\right) & \bar{D}G & 0_{n} & 0_{n} & 0_{n} & -X & *\\ 0_{n} & T_{1}MA-HE_{2} & MD & 0_{n} & 0_{n} & 0_{n} & 0_{n} & -W \end{array}\right]<0$$By substituting Y=KG, H=ML, $S=\gamma X^{-1},$ and $P=\gamma W^{-1}$ into Equation (37) and multiplying everything from the left by diag [$G{-}^{T},\;I,\;I,\;I$, $W^{-1},\;I,\;I,\;I$] and from the right by diag [$G{-}^{T},\;I,\;I,\;I$, $W^{-1},\;I,\;I,\;I$], we obtain the following:(38)$$\left[\begin{array}{cccccccc} -\frac{1}{\gamma }S & * & * & * & * & * & * & *\\ R^{\frac{1}{2}}K & -\gamma I & * & * & * & * & * & *\\ Q^{\frac{1}{2}} & 0_{n} & -\gamma I & * & * & * & * & *\\ 0 & 0_{n} & 0_{n} & -\frac{\mu^{2}I}{\gamma } & * & * & * & *\\ 0 & 0_{n} & 0_{n} & 0_{n} & -\frac{1}{\gamma }P & * & * & *\\ R^{\frac{1}{2}}\left(E_{3}-KE_{1}\right) & 0_{n} & 0_{n} & 0_{n} & 0_{n} & -\gamma I & * & *\\ A+BK & B\left(E_{3}-KE_{1}\right) & \bar{D} & 0_{n} & 0_{n} & 0_{n} & -\gamma S^{-1} & *\\ 0_{n} & T_{1}A-LE_{2} & D & 0_{n} & 0_{n} & 0_{n} & 0_{n} & -\gamma P^{-1} \end{array}\right]<0$$Using the Schur complement, we see that this is equivalent to:(39)$$\left[\begin{array}{ccccc} -S+Q+K^{T}RK & * & * & * & *\\ 0_{n} & -P+{\left(E_{3}-KE_{1}\right)}^{T}R\left(E_{3}-KE_{1}\right) & * & * & *\\ 0_{n} & 0_{n} & -\mu^{2}I & * & *\\ A+BK & B\left(E_{3}-KE_{1}\right) & \bar{D} & -S^{-1} & *\\ 0_{n} & T_{1}A-LE_{2} & D & 0_{n} & P^{-1} \end{array}\right]<0$$Again, applying the Schur complement to Equation (39):(40)$${\left[\begin{array}{cc} A+BK & 0\\ B\left(E_{3}-KE_{1}\right) & T_{1}A-LE_{2}\\ \bar{D} & D \end{array}\right]}^{T}\left[\begin{array}{cc} S & 0\\ 0 & P \end{array}\right]\left[\begin{array}{ccc} A+BK & B\left(E_{3}-KE_{1}\right) & \bar{D}\\ 0 & T_{1}A-LE_{2} & D \end{array}\right]+\left[\begin{array}{ccc} -S+Q+K^{T}RK & * & *\\ 0_{n} & -P+{\left(E_{3}-KE_{1}\right)}^{T}R\left(E_{3}-KE_{1}\right) & *\\ 0_{n} & 0_{n} & -\mu^{2}I \end{array}\right]<0$$
then multiplying the resulting inequality from the left by $\left[x\left(k\right)\;e\left(k\right)\;\bar{\omega }\left(k\right)\right]$ and from the right by $\left[x\left(k\right)\;e\left(k\right)\;\bar{\omega }\left(k\right)\right]$ taking into account Equations (4) and (5), we obtain the following:(41)$$V\left(k+i+1/k\right)-V\left(k+i/k\right)\leq -\left[{\left\|x\left(k+i/k\right)\right\|}_{Q}^{2}+{\|u\left(k+\frac{i}{k})\right\|}_{R}^{2}\right]+\mu^{2}{\left\|\bar{\omega }\left(k\right)\right\|}^{2}$$Consequently, the function V(k+i+1/k) is a strictly decreasing Lyapunov function, which confirms that ω(k)→0, e(k)→0, or x(k)→0 as k→∞. This ends the proof of Theorem 1.At each sampling time, the gain of the virtual observer, obtained by $L=M^{-1}H$, is then used to compute the real observer gain $ℒ=L+\left(T_{1}A-LE_{2}\right)T_{2}$ and to estimate system states and faults.As mentioned earlier in Section 3, the auxiliary variable defined as $\vartheta \left(k\right)=\hat{𝓏}\left(k\right)-T_{2}E_{2}𝓏\left(k\right)$ is used to define the real observer, where:$$\vartheta \left(k+1\right)=\hat{𝓏}\left(k+1\right)-T_{2}E_{2}𝓏\left(k+1\right)$$
$$=T_{1}A\hat{𝓏}\left(k\right)+T_{1}Bu_{c}\left(k\right)+T_{2}E_{2}𝓏\left(k+1\right)+L\left(y_{f}\left(k\right)-E_{2}\hat{𝓏}\left(k\right)\right)-T_{2}E_{2}𝓏\left(k+1\right)$$
(42)$$=(T_{1}A-LE_{2})\hat{𝓏}\left(k\right)+T_{1}Bu_{c}\left(k\right)+Ly_{f}\left(k\right)$$
(43)$$\vartheta \left(k+1\right)=(T_{1}A-LE_{2})\vartheta \left(k\right)+T_{1}Bu_{c}\left(k\right)+\left(L+(T_{1}A-LE_{2}\right))y_{f}\left(k\right)$$This satisfies the form of the designed observer (10) with parameters (11). Finally, we can obtain the estimated values $\hat{𝓏}\left(k\right)=\vartheta \left(k\right)+\bar{C}y_{f}\left(k\right)$. □.

## 5. Robust Stability Analysis

In this section, mathematical formulations are provided to prove that the proposed control law defined in Equation (12) subjected to the constraint Equation (2) robustly stabilizes the close loop system subject to sensor/actuator faults and external disturbances. The provided mathematical formulation, presented in this section, is based on the work of Kothare et al. [34]. Obviously, the optimization problem defined earlier must have a feasible solution at each iteration (sampling time) in order to guarantee the asymptotic stability of the closed-loop system (14). This brings us to propose the following feasibility lemma:

**Lemma** **2.***Feasibility [34]. Any feasible solution found at the sampling instant*k*that satisfies the optimization problem of Theorem 1 is also a feasible solution for all times*t>k*. Therefore, if the targeted optimization problem, discussed earlier in Theorem 1, is feasible at sampling instant*k*, then it is feasible at any sampling instant*t>k.

**Proof.** To prove Lemma 2, it is only necessary to verify that inequality in Equation (18) is feasible for all future measurements of the state and error. x(k+i/k+i)=x(k+i) and e(k+i/k+i)=e(k+i), i≥1 are the future measurements of state and error, respectively. Assuming that the optimization problem in Theorem 1 is feasible at sampling instance k, the following inequality will hold:(44)x(k/k)Sx(k/k)+e(k/k)Pe(k/k)≤γ

The inequality in Equation (44) needs to be feasible for all future measurements in order to prove Lemma 2, which means:(45)x(k+i/k)Sx(k+i/k)+e(k+i/k)Pe(k+i/k)≤γ

At the sampling instant k+1, the values of the state and error are described by x(k+1/k+1)=x(k+1) and e(k+1/k+i)=e(k+1)), respectively. Note that these states and errors are obtained after applying the controller at the sampling instant k, and they must also satisfy the condition in Equation (4); this is written as follows:(46)x(k+1)Sx(k+1)+e(k+1)Pe(k+1)≤γ

This concludes that the feasible solution obtained by Theorem 1 at sampling instant k is also feasible at instant k+1, and hence the problem is feasible at instant k+1. By repeating this argument for all instants, k+2, k+3, …, ∞, we complete the demonstration of Lemma 2.

**Theorem** **2.**
*Robust stability [34]. Assume that Theorem 1 holds at sampling instant k for the system in Equation (14), then the feasible feedback control law obtained from Theorem 1 will robustly asymptotically stabilize the closed-loop system.*


**Proof.** To secure robust asymptotic stability of the closed-loop system, the Lyapunov function $V(k/k)=\chi^{T}(k/k)\psi \chi (k/k)$ must be a strictly decreasing Lyapunov function, where $\chi^{T}(k/k)={\left[x\left(k/k\right)\;e\left(k/k\right)\right]}^{T}$ and $\psi =\left[\begin{array}{cc} S & 0\\ 0 & P \end{array}\right]$.Indicating that ψ(k) and ψ(k+1) are matrices that correspond to the optimal solution at instants k and k+1, respectively, we obtain the following:(47)$$\chi^{T}\left(k+1/k+1\right)\psi \left(k+1\right)\chi \left(k+1/k+1\right)\leq \chi^{T}\left(k+1/k+1\right)\psi \left(k\right)\chi \left(k+1/k+1\right)$$This is obvious, since ψ(k+1) is optimal at the instance *k* + 1, while ψ(k) is only feasible at instant k+1.The controller is said to be a stabilizing controller for all possible states of the system, and the matrix ψ(k) is a Lyapunov matrix if the following inequality holds:(48)$$\chi^{T}\left(k+1/k\right)\psi \left(k\right)\chi \left(k+1/k\right)\leq \chi^{T}\left(k/k\right)\psi \left(k\right)\chi \left(k/k\right)$$If at sampling instant k, the controller, obtained at that sampling instant k, is applied to the system, then the state and error of the system at the next instant k+1, χ(k+1/k+1)=χ(k+1/k), also satisfies the inequality (48). Combining this result with inequality in Equation (47), we obtain the following:(49)$$\chi^{T}\left(k+1/k+1\right)\psi \left(k\right)\chi \left(k+1/k+1\right)\leq \chi^{T}\left(k/k\right)\psi \left(k\right)\chi \left(k/k\right)$$Therefore, V(k/k) is a strictly decreasing Lyapunov function, which confirms that ω(k)→0, e(k)→0, or x(k)→0 as k→∞. □.

## 6. Results and Discussions

In this section, the robustness challenges associated with the observer design are investigated when a discrete-time system is subject to simultaneous input disturbance, actuator faults, and sensor faults. First, a linear mass-spring system with two springs and two masses is modeled and described similar to Equation (1) with the following matrices:$$A=\left[\begin{array}{cccc} 0 & 0 & 1 & 0\\ 0 & 0 & 0 & 1\\ -2 & 1 & -0.5 & 0\\ 2 & -2 & 0 & -1 \end{array}\right],\;B=\left[\begin{array}{c} 0\\ 0\\ 1\\ 0 \end{array}\right],\;C=\left[\begin{array}{cccc} 1 & 0 & 0 & 0\\ 0 & 1 & 0 & 0 \end{array}\right],\;\mathrm{and}\;D=\left[\begin{array}{c} 0.1\\ 0.1\\ 0.1\\ 0.1 \end{array}\right]$$

To investigate the proposed controller, the initial values, sensor and actuator faults, as well as external disturbance are set as follows:x(0)=[0.5, 0, 0.2, 0.1], 𝓏(0)=[0, 0,0,0,0,−0.5,−0.15]
$$\omega \left(k\right)=(sin\pi k+1.7)e^{-0.5k/4},f_{a}\left(k\right)=(sin\pi k)e^{-0.2k}$$
$$f_{s}\left(k\right)=\left[\begin{array}{c} -0.5e^{-0.5k}\\ 0.3(\sin\pi k-0.4)e^{-0.5k} \end{array}\right]$$

Additionally, the MPC parameters are selected as follows: The *Q* and *R* weight matrices of the cost function given in Equation (16) are selected as 100*I* and 10*I*, respectively. The control constraint is considered as ‖u‖≤11,

Based on Theorem 1, the optimization problem provided in Equations (17)–(20) is solved using the YALMIP package [35], while the final observer gain matrix L and controller gain matrix K are:$$K=\left[\begin{array}{cccc} -22.1724 & 1.001 & -18.8971 & -15.1211 \end{array}\right]$$
$$L=\left[\begin{array}{cc} 138.0267 & -82.1369\\ -2.1121 & 12.1524\\ 365.3267 & -220.9021\\ 61.2896 & -24.7623\\ -98.7598 & 77.1905\\ 1.5985 & -2.5398\\ 701.1046 & -399.4967 \end{array}\right]$$

Note that we are dealing with an online LMI optimization problem to minimize the upper bound of the cost function while managing disturbances, sensor/actuator faults, and constraints. The provided gains K and L, shown above, are the obtained gains at the instant *t* = 60 s. However, these two gains are almost similar to the gains at the instant *t* = 40 s when the system states are almost converted to zero. Additionally, the simulation results are presented in Figure 2, Figure 3, Figure 4, Figure 5 and Figure 6.

Figure 2a,b illustrate the states of the system and their estimates when the system is subject to actuator faults, sensor faults, and external disturbance at the same time. The states of the system converge toward zero, which means the proposed controller successfully regulates the system to its equilibrium point.

Additionally, the results shown in Figure 2a,b demonstrate the stability of the closed-loop system under sensor/actuator faults, and external disturbance, while the constructed observer showed an outstanding performance in estimating actuator and sensor faults. It can also be seen that the proposed observers have the ability to track and estimate the nominal states of the system with a minor error and within the first 10 s. Furthermore, the influence of the sensor/actuator is very limited, since the proposed method converges all states to zero within the first 10 s.

This can be verified from Figure 3 where the observer-based predictive control successfully retained the state errors (ex) between nominal model states and faulty states, within an acceptable peak norm (reached a max of 0.1226 within the first 10 s for some states). Notice that these results (estimation error) were achieved under external disturbance, sensor, and actuator faults applied to the system. The synthesized fault-tolerant model predictive controller successfully compensated the system faults with acceptable accuracy.

Figure 4, Figure 5 and Figure 6 illustrate the estimation performance of both sensors and actuator faults. The results indicate that the proposed methodology provided a satisfactory performance in terms of fault estimations. The proposed methodology maintained the estimation error between real faults and estimated faults within an acceptable peak norm (less than 0.078 within the first 10 s). Furthermore, the proposed observer provided superior estimation results of the actuator faults to the observer in [19]. Note that the proposed methodology in [19] only addressed the case of actuator faults, which is completely different from our proposed approach, where both sensor and actuator faults were considered. In conclusion, the proposed fault-tolerant predictive control has the ability to predict and estimate the signal accurately.

## 7. Conclusions

In this work, a robust fault-tolerant model predictive control scheme is proposed to deal with discrete-time systems subject to simultaneous sensor faults, actuator faults, and disturbances with constraints applied to the control input. By augmenting the state model, a virtual observer was first introduced to reduce the fault effects on the system. Then, a real observer was designed, based on the virtual observer, for estimating system states and sensor/actuator faults without the need of using a fault isolation module. Finally, the designed observers were used to establish a robust fault-tolerant predictive control, while Lyapunov theory was used to provide sufficient conditions for stability. Additionally, sufficient stability conditions were formulated as linear matrix inequalities (LMIs) constraints, and the online optimization involved the solution of these LMI constraints. The resulting control law can minimize, at each sampling instant, an upper bound on the robust objective subject input constraints, disturbances, and sensors/actuator faults.

The simulation results show that the proposed methodology can compensate the system faults as well as disturbances with acceptable accuracy and guarantees the robustness and stability of the overall closed-loop system.

## Figures and Tables

**Figure 1 sensors-21-02307-f001:**
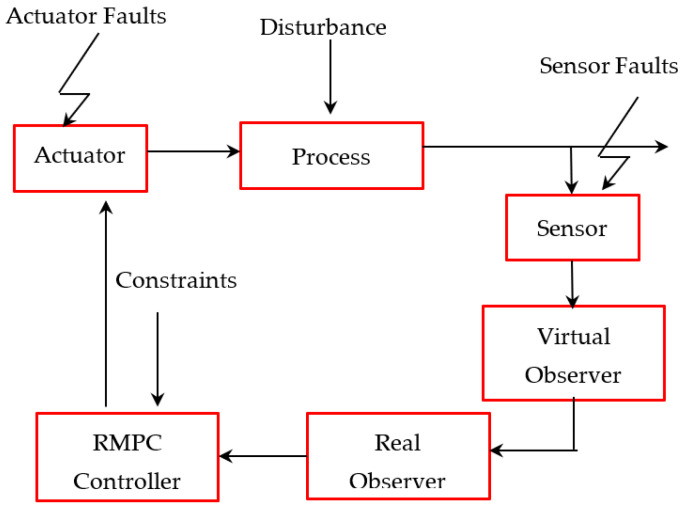
Observer-based robust tolerant predictive control scheme subject to faults and constraints.

**Figure 2 sensors-21-02307-f002:**
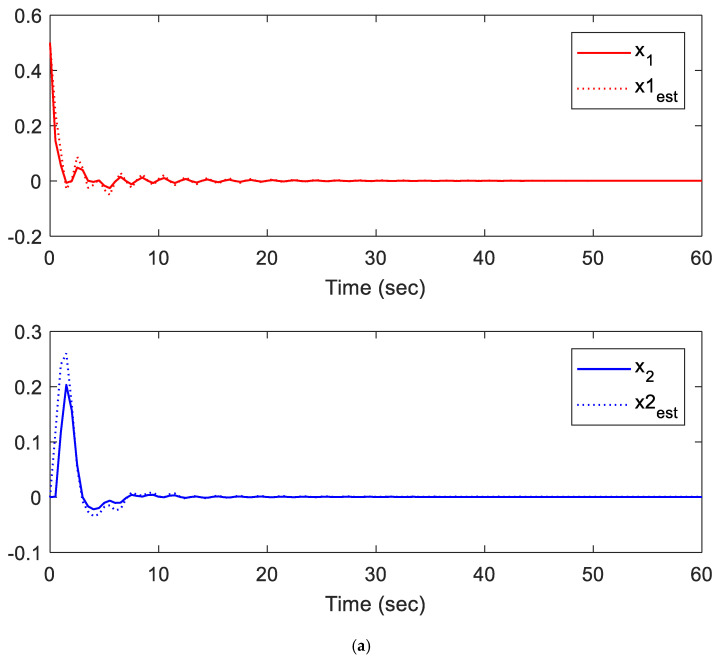
(**a**) The system states *x*_1_ and *x*_2_ subject to faults and their estimates. (**b**) The system states *x*_3_ and *x*_4_ subject to faults and their estimates.

**Figure 3 sensors-21-02307-f003:**
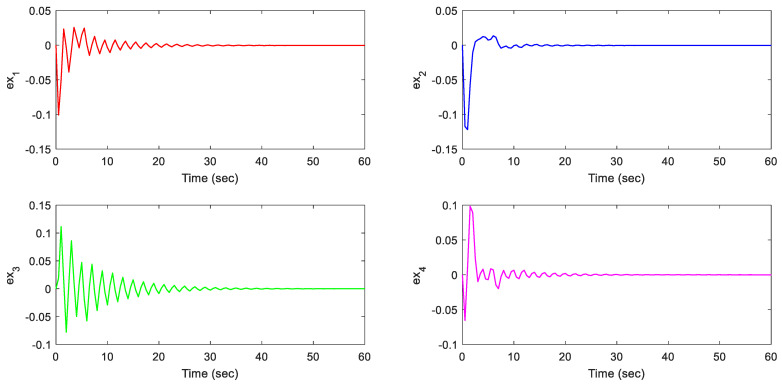
The estimation errors.

**Figure 4 sensors-21-02307-f004:**
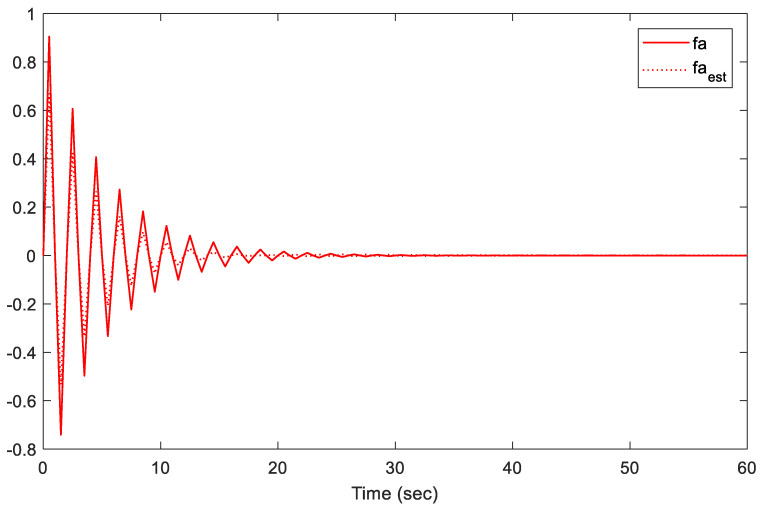
Actuator fault fa and its estimate.

**Figure 5 sensors-21-02307-f005:**
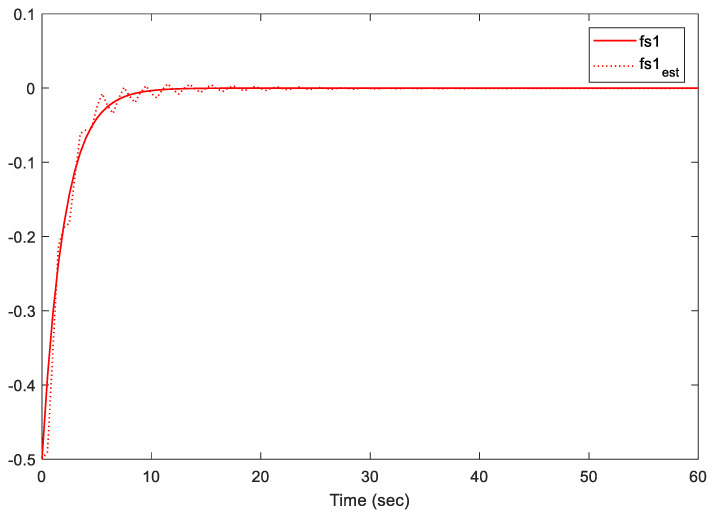
Sensor fault fs1 and its estimate.

**Figure 6 sensors-21-02307-f006:**
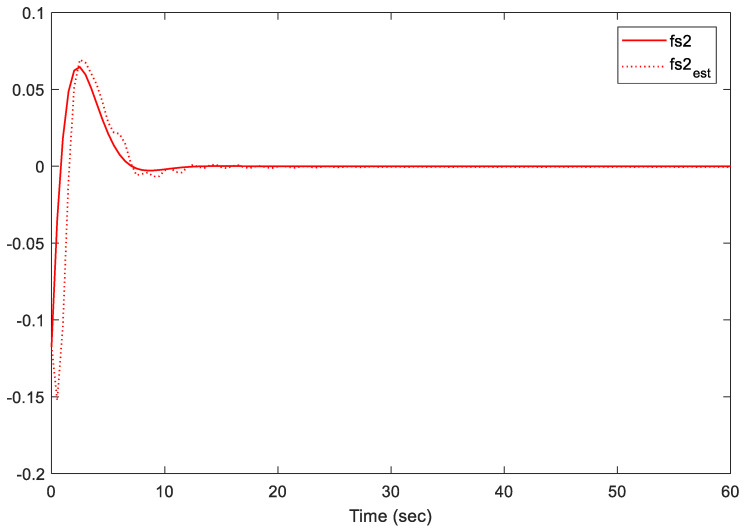
Sensor fault fs2 and its estimate.

## Data Availability

Not applicable.

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
