# Peer review of "A Robust Fault-Tolerant Predictive Control for Discrete-Time Linear Systems Subject to Sensor and Actuator Faults"

_sensors, 2021, doi:10.3390/s21072307_

Round 1

Reviewer 1 Report

1) The abstract can be shortened, in it being necessary to specify only the novelties that the paper brings.
2) The introduction will explain in more detail what the paper is about. Insert more bibliographic references.
3) The methodology should be explained in more detail, including how the proposed theorems, algorithms, and calculation programs were applied. Matrix calculations need also to be explained in more detail.
4) It would be desirable for all six figures presented in the results and discussion section to have a higher resolution to be a little clearer.
5) The conclusions section is too short, it must be much wider, here being finally discussed the results obtained with everything they bring new compared to other similar works. All the novelties presented by the detailed paper must be concluded. They will be briefly presented only in summary.
6) It is good to add more bibliographic references.

Author Response

Please find response attached.

Reviewer 2 Report

Paper present an interesting and well covered in literature topic. For a better readability it is recommended to define all the terms used in equations in the first part of the paper (could be easy to have them in a nomeclature).

System description diagram is recommended to be inserted.

Please specify the provenience of LMI toolbox.

The syntax "demonstrate" the state from line 436 could be improved.

In line 438 please pay attention to capitalization.

Robustness performance indices are missing (eg. sensitivity function).

Paper should be presented in a more coherent manner and results more properly highlighted. The paper is conceived mainly as a theoretical study.

Author Response

Please find response attached.

Reviewer 3 Report

In this paper, the authors have proposed a robust fault-tolerant model predictive control approach for discrete-time linear systems subject to sensor and actuator faults, disturbances, and input constraints. A virtual observer is considered. a real observer is established based on the virtual observer. A robust fault-tolerant model predictive control is synthesized. the efficiency of the proposed controller is tested by simulating a numerical. In my point of view, the paper is on an interesting topic. It is technically sound and mathematically rigorous. However, the following comments are to be addressed. 1- My main concern is about the simultaneous sensor and actuator faults estimation. I agree that most papers have not considered it properly. However, there are some works worth being reviewed. For example, “Hamed Habibi, Ian Howard, Silvio Simani and Afef Fekih, "Decoupling Adaptive Sliding Mode Observer Design for Wind Turbines subject to Simultaneous Faults in Sensors and Actuators," IEEE/CAA J. Autom. Sinica” and https://doi.org/10.1016/j.automatica.2017.12.011 2- What is w in assumption 1? 3- In 8, ?_1??(?) is used which is not available. How have you dealt with this? 4- Please elaborate on How equation 10 is obtained. 5- In equation 9, the term ?_1??(?) + ?_3?_3?(? + 1) is present. While these terms are not available. 6- The stability analysis of the observer error dynamic is not studied. 7- In LMI inequality constraints, usually, the parameters are constants, while, for example, in (16), error e and \bar{x} are contributing. How to guarantee there is a solution? is it implemented online? If so, there is optimization at each time step. 8- Equations number are not consistent 10-b on page 6 while there is no equation 10-a

Author Response

Please find response attached.

Round 2

Reviewer 1 Report

The revised version of the manuscript is OK and suitable now for production, after its last arrangements and check made by the publisher.

Reviewer 2 Report

Paper can be published in its current form.

Reviewer 3 Report

As this manuscript is a revision, I considered the comments and corresponding responses. I believe the authors have addressed all of my comments carefully. Therefore, this paper can be considered for publication.